# When to use parametric models in reinforcement learning?

**Hado van Hasselt**
DeepMind
London, UK
hado@google.com

**Matteo Hessel**
DeepMind
London, UK
mtthss@google.com

**John Aslanides**
DeepMind
London, UK
jaslanides@google.com

## Abstract

We examine the question of when and how parametric models are most useful in reinforcement learning. In particular, we look at commonalities and differences between parametric models and experience replay. Replay-based learning algorithms share important traits with model-based approaches, including the ability to *plan*: to use more computation without additional data to improve predictions and behaviour. We discuss when to expect benefits from either approach, and interpret prior work in this context. We hypothesise that, under suitable conditions, replay-based algorithms should be competitive to or better than model-based algorithms if the model is used only to generate fictional transitions from observed states for an update rule that is otherwise model-free. We validated this hypothesis on Atari 2600 video games. The replay-based algorithm attained state-of-the-art data efficiency, improving over prior results with parametric models. Additionally, we discuss different ways to use models. We show that it can be better to plan *backward* than to plan *forward* when using models to perform credit assignment (e.g., to directly learn a value or policy), even though the latter seems more common. Finally, we argue and demonstrate that it can be beneficial to plan forward for *immediate behaviour*, rather than for credit assignment.

The general setting we consider is learning to make decisions from finite interactions with an environment. Although the distinction is not fully unambiguous, there exist two prototypical families of algorithms: those that learn without an explicit model of the environment (*model free*), and those that first learn a model and then use it to plan a solution (*model based*).

There are good reasons for building the capability to learn some sort of model of the world into artificial agents. Models may allow transfer of knowledge in ways that policies and scalar value predictions do not, and may allow agents to acquire rich knowledge about the world before knowing how this knowledge is best used. In addition, models can be used to *plan*: to use additional computation, without requiring additional experience, to improve the agent's predictions and decisions.

In this paper, we discuss commonalities and differences between *parametric models* and *experience replay* [Lin, 1992]. Although replay-based agents are not always thought of as model-based, replay shares many characteristics that we often associate with parametric models. In particular, we can 'plan' with the experience stored in the replay memory in the sense that we can use additional computation to improve the agent's predictions and policies in between interactions with the real environment.

Our work was partially inspired by recent work by Kaiser et al. [2019], who showed that planning with a parametric model allows for data-efficient learning on several Atari video games. A main comparison was to Rainbow DQN [Hessel et al., 2018a], which uses replay. We explain why their results may perhaps be considered surprising, and show that in a like-for-like comparison Rainbow DQN outperformed the scores of the model-based agent, with less experience and computation.

**Algorithm 1** Model-based reinforcement learning
___
1: Input: state sample procedure $d$
2: Input: model $m$
3: Input: policy $\pi$
4: Input: predictions $v$
5: Input: environment $\mathcal{E}$
6: Get initial state $s \leftarrow \mathcal{E}$
7: **for** iteration $\in \{1, 2, \ldots, K\}$ **do**
8:    **for** interaction $\in \{1, 2, \ldots, M\}$ **do**
9:       Generate action: $a \leftarrow \pi(s)$
10:      Generate reward, next state: $r, s' \leftarrow \mathcal{E}(a)$
11:      $m, d \leftarrow \textsc{UpdateModel}(s, a, r, s')$
12:      $\pi, v \leftarrow \textsc{UpdateAgent}(s, a, r, s')$
13:      Update current state: $s \leftarrow s'$
14:    **end for**
15:    **for** planning step $\in \{1, 2, \ldots, P\}$ **do**
16:      Generate state, action $\tilde{s}, \tilde{a} \leftarrow d$
17:      Generate reward, next state: $\tilde{r}, \tilde{s}' \leftarrow m(\tilde{s}, \tilde{a})$
18:      $\pi, v \leftarrow \textsc{UpdateAgent}(\tilde{s}, \tilde{a}, \tilde{r}, \tilde{s}')$
19:    **end for**
20: **end for**
___

We discuss this in the context of a broad discussion of parametric models and experience replay. We examine equivalences between them, potential failure modes of planning with parametric models, and how to exploit parametric models in addition to, or instead of, using them to provide imagined experiences to an otherwise model-free algorithm.

In particular, we will discuss three different ways to use a learnt, imperfect, model. First, we can plan *forward* for *credit assignment*. This means we roll the model forward from real states, for instance stored in a replay buffer, and use the resulting modelled transitions to learn predictions or policies. We argue that this can be worse than planning *backward* from real states, because the former will involve updating real states with fictional experiences, whereas the latter only involves updating fictional states, which seems safer. This hypothesis is validated empirically. Finally, a third use of a model is to plan forward from the current state, to help determine the *immediate behaviour*.

We believe that planning backward for credit assignment or planning forward for behaviour may be more beneficial than planning forward for credit assignment. To see why, consider an inaccurate model that for instance predicts a transition to some magical world that is not truly there, thereby providing a fictional path to high rewards. Planning forward for credit assignment may then result in incorrect predictions and policies that assume this fiction is real. Instead, planning backward when the model is inaccurate may lead to updates to fictional states that are unreachable, which may or may not be useful, but is less likely to be harmful than updating real states with fictional transitions. However, if we do plan forward but only use the inaccurate model to inform our behaviour rather than trusting its transitions as if they are real, then we might expect an agent to go and see whether there is in fact a magical world around the corner. This may result in useful data, and perhaps even useful exploration, regardless of whether the modelled transition in fact exists or not.

## 1 Model-based reinforcement learning

We now define the terminology that we use in the paper, and present a generic algorithm that encompasses both model-based and replay-based algorithms.

We consider the reinforcement learning setting [Sutton and Barto, 2018] in which an *agent* interacts with an *environment*, in the sense that the agent outputs actions and then obtains observations and rewards from the environment. We consider the *control* setting, in which the goal is to optimise the accumulation of the rewards over time by picking appropriate sequences of actions. The action an agent outputs typically depends on its *state*. This state is a function of past observations; in some cases it is sufficient to just use the immediate observation as state, in other cases a more sophisticated agent state is required to yield suitable decisions. The state of the agent should not be confused with

the state of the environment, which is typically not fully observable to the agent, and is also typically much too large to reason about directly.

We use the word *planning* to refer to any algorithm that uses additional computation to improve its predictions or behaviour without consuming additional data. Conversely, we reserve the term *learning* for updates that depend on newly observed experience.

The term *model* will refer to functions that take a state and action as input, and that output a reward and next state. Sometimes we may have a perfect model, as in board games (e.g., chess and go); sometimes the model needs to be learnt before it can be used. Models can be stochastic, to approximate inherently stochastic transition dynamics, or to model the agent's uncertainty about the future. Expectation models are deterministic, and output (an approximation of) the expected reward and state. If the true dynamics are stochastic, iterating expectation models multiple steps may be unhelpful, as an expected state may itself not be a valid state; the output of a model may not have useful semantics when using an expected state as input rather than a real state [cf. Wan et al., 2019]. Planning is associated with models, because a common way to use computation to improve predictions and policies is to search using a model. For instance, in Dyna [Sutton, 1990], learning and planning are combined by using new experience to learn both the model and the agent's predictions, and then planning to further improve the predictions.

*Experience replay* [Lin, 1992] refers to storing previously observed transitions to replay later for additional updates to the predictions and policy. Replay may be used for planning and, when queried at state-action pairs we have observed, experience replay may be indistinguishable from an accurate model. Sometimes, there may be no practical differences between replay and models, depending on how they are used. On the other hand, a replay memory is less flexible than a model, since we cannot query it at arbitrary states that are not present in the replay memory.

## 1.1 A generic algorithm

Algorithm 1 is a generic model-based learning algorithm. It runs for $K$ iterations, in each of which $M$ interactions with the environment occur. The total number of interactions is thus $T \equiv K \times M$. The experience is used to update a model (line 11) and the policy or predictions of the agent (line 12). Then, $P$ steps of planning are performed, where transitions sampled from the model are used to update the agent (line 18). For $P = 0$, the model is not used, hence the algorithm is model-free (we could then also skip line 11). If $P > 0$, and the agent update in line 12 does not do anything, we have a purely model-based algorithm. The agent updates in lines 12 and 18 could differ, or they could treat real and modelled transitions equivalently.

Many known algorithms from the model-based literature are instances of algorithm 1. If lines 12 and 18 both update the agent's predictions in the same way, the resulting algorithm is known as Dyna [Sutton, 1990] – for instance, if predictions $v$ include action values (normally denoted with $q$) and we update using Q-learning [Watkins, 1989, Watkins and Dayan, 1992], we obtain Dyna-Q [Sutton and Barto, 2018]. One can extend Algorithm 1 further, for instance by allowing planning and model-free learning to happen simultaneously. Such extensions are orthogonal to our discussion and we do not discuss them further.

Some algorithms typically thought of as being model-free also fit into this framework. For instance, DQN [Mnih et al., 2013, 2015] and neural-fitted Q-iteration [Riedmiller, 2005] match Algorithm 1, if we stretch the definitions of 'model' to include the more limited replay buffers. DQN learns from transitions sampled from a replay buffer by using Q-learning with neural networks. In Algorithm 1, this corresponds to updating a non-parametric model, in line 11, by storing observed transitions in the buffer (perhaps overwriting old transitions); line 17 then retrieves a transition from this buffer. The policy is only updated with transitions sampled from the replay buffer (i.e., line 12 has no effect).

## 2 Model properties

A main advantage of using models is the ability to *plan*: to use additional computation, but no new data, to improve the agent's policy or predictions. Sutton and Barto [2018] illustrate the benefits of planning in a simple grid world (Figure 1, on the left), where the agent must learn to navigate along the shortest path to a fixed goal location. On the right of Figure 1 we use this domain to show how the performance of a replay-based Q-learning agent (**blue**) and that of a Dyna-Q agent

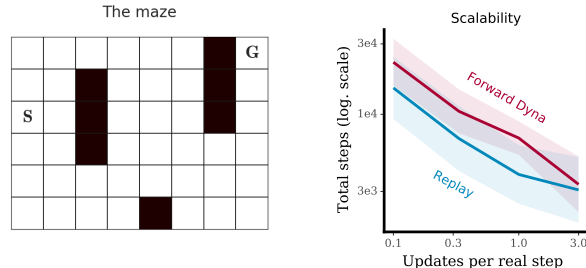

Figure 1: **Left**: the layout of the grid world [Sutton and Barto, 2018], 'S' and 'G' denote the start and goal state, respectively. **Right**: Q-learning with replay (blue) or Dyna-Q with a parametric model (red); $y$-axis: the total number of steps to complete 25 episodes of experience, $x$-axis: the number of updates per step in the environment. Both axes are on a logarithmic scale.

(**red**) scale similarly with the amount of planning (measured in terms of the number of updates per real environment step). Both agents use a multi-layer perceptron to approximate action values, but Dyna-Q also used identical networks to model transitions, terminations and rewards. The algorithm is called 'forward Dyna' in the figure, because it samples states from the replay and then steps forward one step using the model. Later we will consider a variant that, instead, steps backward with an inverse model. The appendix contains further details on the experiments.

## 2.1 Computational properties

There are clear computational differences between using parametric models and replay. For instance, Kaiser et al. [2019] use a fairly large deep neural network to model the pixel dynamics in Atari, which means predicting a single transition can require non-trivial computation. In general, parametric models typically require more computations than it takes to sample from a replay buffer.

On the other hand, replay tightly couples model capacity and memory requirements: each transition that is stored takes up a certain amount of memory. If we do not remove any transitions, the memory can grow unbounded. If we limit the memory usage, then this implies that the effective capacity of the replay is limited as any transitions we replace are forgotten completely. In contrast, parametric models may be able to achieve good accuracy with a fixed and comparatively small memory footprint.

## 2.2 Equivalences

Suppose we manage to learn a model that perfectly matches the transitions observed thus far. If we would then use such a perfect model to generate experiences only from states that were actually observed, the resulting updates would be indistinguishable from doing experience replay. In that sense, replay matches a perfect model, albeit only from the states we have observed.[1] Therefore, all else being equal, we would expect that using an imperfect (e.g., parametric) model to generate fictional experiences from truly observed states should probably not result in better learning.

There are some subtleties to this argument. First, the argument can be made even stronger in some cases. When making linear predictions with least-squares temporal-difference learning [LSTD, Bradtke and Barto, 1996, Boyan, 1999], the model-free algorithm on the original data does not require (or indeed benefit from) planning: the solution will already be a best fit (in a least squares sense) even with a single pass through the data. In fact, if we fit a linear model to the data and then fully solve this model, the solution is equal to the LSTD solution [Parr et al., 2008]. One can also show that exhaustive replay with linear TD($\lambda$) [Sutton, 1988] is equivalent to a one-time pass through the data with LSTD($\lambda$) [van Seijen and Sutton, 2015], because replay similarly allows us to solve the empirical 'model' that is implicitly defined by the observed data.

These full equivalences are however limited to linear prediction, and do not extend straightforwardly to non-linear functions, or to the control setting. This leaves open the question of when to use a parametric model rather than replay, or vice versa.

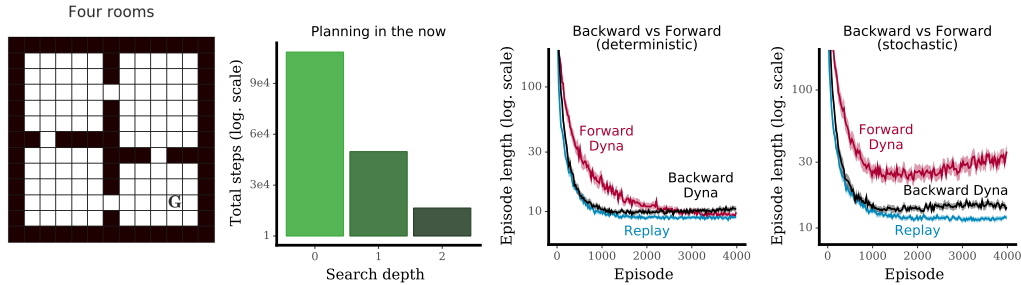

Figure 2: **Left**: four rooms grid world [Sutton et al., 1998]. **Center-left**: planning forward from the current state to update the current behaviour (0 steps corresponds to Q-learning); $y$-axis: total number of steps required to complete 100 episodes, $x$-axis: search depth. **Center-right**: comparing replay (blue), forward Dyna (red), and backward Dyna (black); $y$-axis: episode length (logarithmic scale), $x$-axis: number of episodes. **Right**: adding stochasticity to the transition dynamics (in the form of a 20% probability of transitioning to a random adjacent cell irrespectively of the action), then comparing again replay (blue), forward Dyna (red), and backward Dyna (black); $y$-axis: episode length (logarithmic scale), $x$-axis: number of episodes

## 2.3   When do parametric models help learning?

When should we expect benefits from learning and using a parametric model, rather than using the actual data? We discussed important computational differences above. Here we focus on learning efficiency: when do parametric models help learning?

First, parametric models may be useful to plan into the future to help determine our policy of behaviour. The ability to generalise to unseen or counter-factual transitions can be used to plan from the *current state* into the future (sometimes called planning 'in the now' [Kaelbling and Lozano-Pérez, 2010]), even if this exact state has never before been observed. This is commonly and successfully employed in model-predictive control [Richalet et al., 1978, Morari and Lee, 1999, Mayne, 2014, Wagener et al., 2019]. Classically, the model is constructed by hand rather than learnt directly from experience, but the principle of planning forward to find suitable behaviour is the same. It is not possible to replicate this with standard replay, because in interesting rich domains the current state will typically not exactly appear in the replay. Even if it would, replay does not allow easy generation of *possible* next states, in addition to the one trajectory that actually happened.

If we use a model to select actions, rather than trusting its imagined transitions to update the policy or predictions, it may be less essential to have a highly accurate model. For instance, the model may predict a shortcut that does not actually exist; using this to then steer behaviour results in experience that is both suitable to correct the error in the model, and that yields the kind of directed, temporally consistent behaviour typically sought for exploration purposes [Lowrey et al., 2019].

We illustrate this with an experiment on a classic four room grid-world [Sutton et al., 1998]. We learnt a tabular forward model that generates transitions $(s, a) \rightarrow (r, \gamma, s')$, where $s$ and $s'$ are states, $a$ is an action, $r$ is a reward, and $\gamma \in [0, 1]$ is a discount factor. We then used this model to plan via a simple breadth-first search up to a fixed depth, bootstrapping from a value function $q(s, a)$ learnt via standard Q-learning. We then use the resulting planned values of the actions at the current state to behave. This process can be interpreted as using a multi-step greedy policy [Efroni et al., 2018] to determine behaviour, instead of the more standard one-step greedy policy. The results are illustrated in the second plot in Figure 2: more planning was beneficial.

In addition to planning forward to improve behaviour, models may be useful for credit assignment through *backward* planning. Consider an algorithm where, as before, we sample real visited states from a replay buffer, but instead of planning one step into the future from these states we plan one step backward. One motivation is that if the model is poor then planning a step forward will update the real sampled state with a misleading imagined transition. This will potentially cause harmful updates to the value at these real states. Conversely, if we plan backwards we update an imagined state. If the model is poor this imagined state perhaps does not resemble any real state. Updating such fictional states seems less harmful. When the model becomes very accurate, forward and backward planning both start to be equally useful. For a purely data-driven (partial) model, such as a replay buffer, there

is no meaningful distinction. But with a learnt model that is at times inaccurate, backward planning may be less error-prone than forward planning for credit assignment.

We illustrate potential benefits of backward planning with a simple experiment on the four-room environment. In the two right-most plots of Figure 2, we compare the performance of applying tabular Q-learning to transitions generated by a forward model (**red**), a backward model (**black**), or replay (**blue**). The forward model learns distributions over states, rewards, and terminations $\Pr(r, \gamma, s'|s, a)$. The backward model learns the inverse $\Pr(s, a|r, \gamma, s')$. Both use a Dirichlet(1) prior. We evaluated the algorithms in the deterministic four-room environment, as well as in a stochastic variant where on each step there is a 20% probability of transitioning to a random adjacent cell irrespective of the action. In both cases, backward planning resulted in faster learning than forward planning. In the deterministic case, the forward model catches up later in learning, reaching the same performance of replay after 2000 episodes; instead, planning with a backward model is competitive with replay in early learning but performs slightly worse later in training. We conjecture that the slower convergence in later stages of training may be due to the fact that predicting the source state and action in a transition is a non-stationary problem (as it depends on the agent's policy), and given that early episodes include many more transitions than later ones, it can take many episodes for a Bayesian model to forget policies observed early in training. The lack of convergence to the optimal policy for the forward planning algorithm in the stochastic setting may be due to the independent sampling of the successor state and reward, which may result in inconsistent transitions. Both these issues may be addressed by a suitable choice of the model. More detailed investigations are out of scope for this paper, but it is good to recognise that such modelling choices have measurable effects on learning.

## 3 A failure to learn

We now describe how planning in a Dyna-style learning algorithm can, perhaps surprisingly easily, lead to catastrophic learning updates.

Algorithms that combine function approximation (e.g., neural networks), bootstrapping (as in temporal difference methods [Sutton, 1988]), and off-policy learning [Sutton and Barto, 2018, Precup et al., 2000] can be unstable [Williams and Baird III, 1993, Baird, 1995, Sutton, 1995, Tsitsiklis and Van Roy, 1997, Sutton et al., 2009, 2016] — this is sometimes called the *deadly triad* [Sutton and Barto, 2018, van Hasselt et al., 2018].

This has implications for Dyna-style learning, as well as for replay methods [cf. van Hasselt et al., 2018]. When using replay it is sometimes relatively straightforward to determine how off-policy the state sampling distribution is, and the sampled transitions will always be real transitions under that distribution (assuming the transition dynamics are stationary). In contrast, the projected states given by a parametric model may differ from the states that would occur under the real dynamics, due to modelling error. The update rule will then be solving a predictive question for the MDP induced by the model, but with a state distribution that does not match the on-policy distribution in that MDP.

To understand this issue better, consider using Algorithm 1 to estimate expected cumulative discounted rewards $v_\pi(s) = \mathbb{E}[R_{t+1} + \gamma R_{t+2} + \ldots \mid S_t = s, \pi]$ for a policy $\pi$ by updating $v_{\mathbf{w}}(s) \approx v_\pi(s)$ with temporal difference (TD) learning [Sutton, 1988]:

$$\mathbf{w} \leftarrow \mathbf{w} + \alpha \delta_t \nabla_{\mathbf{w}} v_{\mathbf{w}}(S_t), \qquad \text{with} \qquad \delta_t \equiv R_{t+1} + \gamma_{t+1} v_{\mathbf{w}}(S_{t+1}) - v_{\mathbf{w}}(S_t), \qquad (1)$$

where $R_{t+1} \in \mathbb{R}$ and $\gamma_{t+1} \in [0, 1]$ are the reward and discount on the transition from $S_t$ to $S_{t+1}$, and $\alpha > 0$ is a small step size. Consider linear predictions $v_{\mathbf{w}}(S_t) = \mathbf{w}^\top \mathbf{x}_t \approx v_\pi(S_t)$, where $\mathbf{x}_t \equiv \mathbf{x}(S_t)$ is a feature vector for state $S_t$. The expected TD update is then $\mathbf{w} \leftarrow (\mathbf{I} - \alpha \mathbf{A})\mathbf{w} + \alpha \mathbf{b}$, with $\mathbf{b} = \mathbb{E}[R_{t+1}\mathbf{x}_t]$ and $\mathbf{A} = \mathbb{E}[\mathbf{x}_t \mathbf{x}_t^\top - \gamma \mathbf{x}_t \mathbf{x}_{t+1}] = \mathbf{X}^\top \mathbf{D}(\mathbf{I} - \gamma \mathbf{P}^\top)\mathbf{X}$, where the expectation is over the transition dynamics and over the sampling distribution $d$ of the states. The transition dynamics can be written as a matrix $\mathbf{P}$, that contains the probabilities $[\mathbf{P}]_{ij} = p(S_{t+1} = i \mid S_t = j, \pi)$ of transitioning from any state $j$ to any state $i$ under policy $\pi$. The diagonal matrix $\mathbf{D}$ contains the probabilities $[\mathbf{D}]_{ii} = d(i) = P(S_t = i \mid \pi)$ of sampling each state $i$ on its diagonal. The matrix $\mathbf{X}$ contains the feature vectors $\mathbf{x}(s)$ of all states on its rows, and maps between state and feature space. Note that both $\mathbf{P}$ and $\mathbf{D}$ are linear operators in state space, not feature space.

These updates are guaranteed to be stable (i.e., converge) if $\mathbf{A} = \mathbf{X}^\top \mathbf{D}(\mathbf{I} - \gamma \mathbf{P}^\top)\mathbf{X}$ is positive semi-definite [Sutton et al., 2016], with spectral radius $\rho(\mathbf{A})$ smaller than $1/\alpha$. The deadly triad occurs when $\mathbf{D}$ and $\mathbf{P}$ do not match: then $\mathbf{A}$ can be negative definite, the spectral radius $\rho(\mathbf{I} - \alpha \mathbf{A})$

can be larger than one, and the weights can diverge. This can happen when $\mathbf{D}$ does not correspond to the steady-state distribution of the policy that conditions $\mathbf{P}$ — that is, if we update off-policy.

**Proposition 1.** *Consider uniformly replaying transitions from a buffer containing full episodes (e.g., add new full episodes on termination, potentially remove an old full episode), and using these transitions in the TD algorithm defined by update* (1). *This algorithm is stable.*

*Proof.* The replay buffer defines an empirical model, where the induced policy is the empirical distribution of actions: $\tilde{\pi}(a|s) = n(s,a)/n(s)$, where $n(s)$ and $n(s,a)$ are the number of times $s$ and the pair $(s,a)$ show up in the replay. (The behaviour policy can change while filling the replay, the resulting empirical policy is then a sample of a mixture of these policies). The empirical transitions $[\tilde{\mathbf{P}}]_{ij} = n(i,j)/n(i)$ and state distributions $[\tilde{\mathbf{D}}]_{ii} = n(s)/N$, where $N$ is the total size of the replay buffer, then both correspond to the same empirical policy. Therefore, $\rho(\tilde{\mathbf{X}}^\top \tilde{\mathbf{D}}(\mathbf{I} - \gamma \tilde{\mathbf{P}}^\top)\tilde{\mathbf{X}}) > 0$, and TD will be stable and will not diverge. $\qquad\square$

This proposition can be extended to the case where transitions are added to the replay one at the time, rather then in full episodes. If, however, we sample states according to a non-uniform distribution (e.g., using prioritised replay) this can make replay-based algorithms less stable and potentially divergent [cf. van Hasselt et al., 2018].

We now show that a very similar algorithm that uses models in place of replay can diverge.

**Proposition 2.** *Consider uniformly replaying states from a replay buffer, then generating transitions with a learnt model $\hat{p}_m$, and using these transitions in a TD update* (1). *This algorithm can diverge.*

*Proof.* The learnt dynamics $\hat{\mathbf{P}}_m \approx \mathbf{P}$ do not necessarily match the empirical dynamics of the replay, which means that the empirical replay distribution $d$, used in the updates, does not necessarily correspond to the steady-state distribution of these dynamics. Then the model error could lead to a negative definite $\hat{\mathbf{A}} \equiv \mathbf{X}^\top \tilde{\mathbf{D}}(\mathbf{I} - \gamma \hat{\mathbf{P}}_m^\top)\mathbf{X}$, resulting in a spectral radius $\rho(I - \alpha\hat{\mathbf{A}}) > 1$, and divergence of the parameters $\mathbf{w}$. $\qquad\square$

Intuitively, the issue is that the model $m$ can lead to states that are uncommon, or impossible, under the sampling distribution $d$. Those states are not sampled to be updated directly, but do change through generalisation when sampled states are updated. This can lead to divergent learning dynamics.

There are ways to mitigate the failure described above. First, we could repeatedly iterate the model, and sample transitions *from* the states the model generates as well as *to* those states, to induce a state distribution that is consistent with the model. This is not fully satisfactory, as states typically become ever-more unrealistic when iterating a learnt model, although there is some indication this may be helpful [Holland et al., 2018]. Second, we could rely less on bootstrapping by using multi-step returns [Sutton, 1988, van Hasselt and Sutton, 2015, Sutton and Barto, 2018]. This mitigates the instability [cf. van Hasselt et al., 2018]. In the extreme, full Monte-Carlo updates do not diverge, though they would have high variance. Third, we could employ algorithms specifically for stable off-policy learning, although these are often specific to the linear setting [Sutton et al., 2008, 2009, van Hasselt et al., 2014] or assume the sampling is done on trajectory [Sutton et al., 2016]. Note that several algorithms exist that correct the *return* towards a desired policy [Harutyunyan et al., 2016, Munos et al., 2016], which is a separate issue from off-policy sampling of *states*. Although off-policy learning algorithms may be part of the long-term answer, we do not yet have a definitive solution. To quote Sutton and Barto [2018]: *The potential for off-policy learning remains tantalising, the best way to achieve it still a mystery.*

Understanding such failures to learn is important to understand and improve our algorithms. However, just because divergence *can* occur does not mean it *does* occur [cf. van Hasselt et al., 2018]. Indeed, in the next section we compare a replay-based algorithm to a model-based algorithm which was stable enough to achieve impressive sample-efficiency on the Atari benchmark.

## 4 Model-based algorithms at scale

We now discuss two algorithms in more detail: first SimPLe [Kaiser et al., 2019], which uses a parametric model, then Rainbow DQN [Hessel et al., 2018a], which uses experience replay (and was used as baseline by Kaiser et al.).

**SimPLe**  Kaiser et al. [2019] showed data-efficient learning is possible in Atari 2600 videos games from the arcade learning environment [Bellemare et al., 2013] with a purely model-based approach: only updating the policy with data sampled from a learnt parametric model $m$. The resulting "simulated policy learning" (SimPLe) algorithm performed relatively well after just 102,400 interactions (409,600 frames — two hours of simulated play) within each game. In Algorithm 1, this corresponds to setting $K \times M = 16 \times 6400 = 102,400$. Although SimPLe used limited data, it used a large number of samples from the model, similar to using $P = 800,000$.[2]

**Rainbow DQN**  One of the main results by Kaiser et al. [2019] was to compare SimPLe to Rainbow DQN [Hessel et al., 2018a], which combines the DQN algorithm [Mnih et al., 2013, 2015] with double Q-learning [van Hasselt, 2010, van Hasselt et al., 2016], dueling network architectures [Wang et al., 2016], prioritised experience replay [Schaul et al., 2016], noisy networks for exploration [Fortunato et al., 2017], and distributional reinforcement learning [Bellemare et al., 2017]. Like DQN, Rainbow DQN uses mini-batches of transitions sampled from experience replay [Lin, 1992] and uses Q-learning [Watkins, 1989] to learn the action-value estimates which determine the policy. Rainbow DQN uses multi-step returns [cf. Sutton, 1988, Sutton and Barto, 2018] rather than the one-step return used in the original DQN algorithm.

## 4.1  A data efficient Rainbow DQN

In the notation of Algorithm 1, the total number of transitions sampled from replay during learning will be $K \times P$, while the total number of interactions with the environment will be $K \times M$. Originally, in both DQN and Rainbow DQN, a batch of 32 transitions was sampled every 4 real interactions. So $M = 4$ and $P = 32$. The total number of interactions was 50M (200 million frames), which means $K = 50\text{M}/4 = 12.5\text{M}$.

In our experiments below, we trained Rainbow DQN for a total number of real interactions comparable to that of SimPLe, by setting $K = 100,000$, $M = 1$ and $P = 32$. The total number of replayed samples (3.2 million) is then still less than the total number of model samples used in SimPLe (15.2 million). Rainbow DQN is also more efficient computation-wise, since sampling from a replay buffer is faster than generating a transition with a learnt model.

The other changes we made to make Rainbow DQN more data efficient were to increase the number of steps in the multi-step returns from 3 to 20, and to reduce the number of steps before we start sampling from replay from $20,000$ to $1600$. We used the fairly standard convolutional Q network from Hessel et al. [2018b]. We have not tried to exhaustively tune the algorithm and we do not doubt that the algorithm can be made even more data efficient by futher tuning its hyper-parameters.

## 4.2  Empirical results

We ran Rainbow DQN on the same 26 Atari games reported by Kaiser et al. [2019]. In Figure 3, we plotted the performance of our version of Rainbow DQN as a function of the number of interactions with the environment. Performance was measured in terms of episode returns, normalised using human and random scores [van Hasselt et al., 2016], and then aggregated across the 26 games by taking their median. Error bars are shown as computed over the 5 independent replicas of each experiment. The final performance of SimPLe, according to the same metric, is shown in Figure 3 as a dashed horizontal line.

As expected, the hyper-parameters proposed by Hessel et al. [2018a] for the larger-data regime of 50 million interactions are not well suited to a regime of extreme data-efficiency (purple line in Figure 3). Performance was better for our slightly-tweaked data-efficient version of Rainbow DQN (red), that matched the performance of SimPLe after just 70,000 interactions with the environment, reaching roughly 25% higher performance by 100,000 interactions. The performance of our agent was superior to that of SimPLe in 17 out of 26 games. More detailed results are included in the appendix, including ablations and per-game performance.

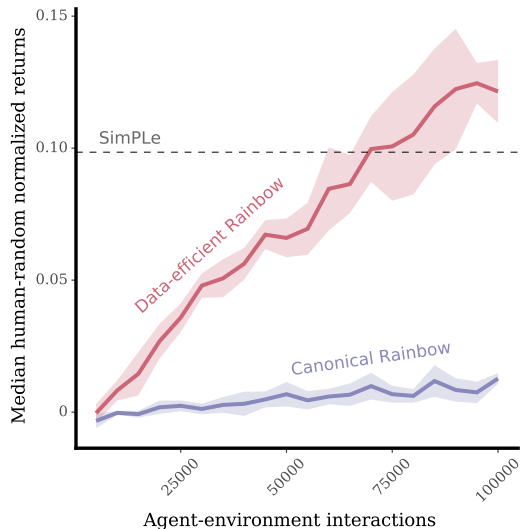

Figure 3: Median human-normalised episode returns of a tuned Rainbow, as a function of environment interactions (=frames/action repeats). The horizontal dashed line corresponds to the performance of SimPLe [Kaiser et al., 2019]. Error bars are computed over 5 seeds.

# 5 Conclusions

We discussed commonalities and differences between replay and model-based methods. In particular, we discussed how model errors may cause issues when we use a parametric model in a replay-like setting, where we sample observed states from the past. We note that model-based learning can be unstable in theory, and hypothesised that replay is likely a better strategy under that state sampling distribution. This is confirmed by at-scale experiments on Atari 2600 video games, where our replay-based agent attained state-of-the-art data efficiency, besting the impressive model-based results by Kaiser et al. [2019].

We further hypothesised that parametric models are perhaps more useful when used either 1) to plan backward for credit assignment, or 2) to plan forward for behaviour. Planning forward for credit assignment was hypothesised and shown to be less effective, even though the approach is quite common. The intuitive reasoning was that when the model is inaccurate, then planning backwards with a learnt model may lead to updating fictional states, which seems less harmful than updating real states with inaccurate transitions as would happen in forward planning for credit assignment. Forward planning for *behaviour*, rather than credit assignment, was deemed potentially useful and less likely to be harmful for learning, because the resulting plan is not trusted as real experience by the prediction or policy updates. Empirical results supported these conclusions.

There is a rich literature on model-based reinforcement learning, and this paper cannot cover all the potential ways to plan with learnt models. One notable topic that is out of scope for this paper is the consideration of abstract models [Silver et al., 2017] and alternative ways to use these models in addition to classic planning [cf. Weber et al., 2017].

Finally, we note that our discussion focused mostly on the distinction between parametric models and replay, because these are the most common, but it is good to acknowledge that one can also consider *non-parametric* models. For instance, one could apply a nearest-neighbours or kernel approach to a replay buffer, and thereby obtain a non-parametric model that can be equivalent to replay when sampled at the observed states, but that can interpolate and generalise to unseen states when sampled at other states [Pan et al., 2018]. This is conceptually an appealing alternative, although it comes with practical algorithmic questions of how best to define distance metrics in high-dimensional state spaces. This seems another interesting potential avenue for more future work.

**Acknowledgments**

The authors benefitted greatly from feedback from Tom Schaul, Adam White, Brian Tanner, Richard Sutton, Theophane Weber, Arthur Guez, and Lars Buesing.

## Footnotes

[1]One could go one step further and extend replay full non-parametric models. For instance Pan et al. [2018] use kernel methods to allow querying the replay-based model at states that are not stored in the buffer.

[2]The actual number of reported model samples was $19 \times 800,000 = 15.2$ million, because $P$ was varied depending on the iteration.

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
