[Supplementary Material · NeurIPS_model_based_RL_with_appendix.pdf]

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

# Appendix

## A  Divergence example

As a concrete illustration of the issue discussed in Section 3, consider the two-state Markov reward process (MRP) depicted in Figure 4a. This example is similar in nature to other examples from the literature [Baird, 1995, Tsitsiklis and Van Roy, 1997]. On each transition with probability $p$ we transition to state $s = 1$ (left), and with probability of $1 - p$ transition to $s = 2$ (right). All rewards are 0, discount is $\gamma = 0.99$. Each state has a single feature $x(s) = s$. The goal is to learn a weight $w$ such that $v_{\mathbf{w}}(s) = w \times x(s)$ is accurate. The optimal weight is, trivially, $w = 0$.

As discussed, the expected update can diverge if the sampling distribution of states $d$ does not match the sampling distribution under model $m$. Figure 4b shows under which sampling probabilities $d(s) = P(S_t = s)$ and transition probabilities $P(S_{t+1} = 1|S_t)$ the updates diverge. Divergence occurs when the probability of sampling state $s = 1$ (under $d$) is sufficiently higher than the transition probability into state $s = 1$. Note how oversampling state $s = 2$ is less harmful for this specific choice of function approximation.

Updates do not diverge because the learnt model is inaccurate, but because of a mismatch between the model dynamics and the state sampling distribution. Divergence can thus occur even when using the true dynamics, if $d$ does not match the steady-state distribution induced by such dynamics. For a true dynamics of $p(S_{t+1} = 1|S_t) = 0.5$, Figure 4c shows the likelihood of observing divergence as a function of the number of samples used to estimate the empirical distribution $d$, assuming a perfect model and unbiased data-dependent estimates of $d$.

## B  Experiment details: Scalability of planning

The layout of maze used in these experiments is shown in the main text. The agent can see a $5 \times 5$ portion of the maze, centered in its current location, where walls are encoded with 1s, and free cells as 0s. The agent can choose among 4 actions (up, down, left, right) that result in deterministic transitions to the adjacent cell, as long as such cell is empty; if the cell is a wall, the action has no effect.

Both the forward Dyna agent and the replay-based Q-learning agent used a multi-layer perceptron (with two fully connected hidden layers, of size 20, and ReLU activations throughout) to approximate Q-values. The final output layer had no activation, and had only 4 nodes, one per action. The forward Dyna agent used separate networks with the same hidden layers to model state transitions, rewards and terminations; the output layers of these had 25, 1, and 1 outputs. Both agents use a replay with a capacity of 10000 transitions; the Q-networks are updated with double Q-learning, on mini-batches of size 32; updates are rescaled by TensorFlow's implementation of the Adam optimizer, using a learning rate of $1e - 3$. In the replay-based agent the update is computed using only the real data from the transition $s_{t-1}, a_{t-1}, r_t, \gamma_t, s_t$; in forward dyna the fictional transition $s_{t-1}, a_{t-1}, m_R(s_{t-1}, a_{t-1}), m_T(s_{t-1}, a_{t-1}), m_S(s_{t-1}, a_{t-1})$ is used instead, where $m_R, m_T, m_S$ are the outputs of the three neural networks used to parameterize the model.

## C  Experiment details: Benefits of Planning

For these experiments we run on the four-rooms environment shown in the text. At the beginning of each episode, the agent's starting position is randomized and the goal position is held fixed. The dynamics are deterministic, with four actions that move the agent in the four cardinal directions, and a no-op action. The state is fully-observed, and we use a tabular (state-index) representation for these experiments. In both experiments we learn an exact Bayesian tabular model. We also learn a tabular value function in tandem using one-step (tabular) Q-learning.

## D  Additional results on Atari

In Figure 5a and 5b we show the results of ablation experiments performed to isolate the effect of increasing the bootstrapping parameter and the effect of increasing the frequency of updates. In Figure 5a we show the effect of varying the bootstrapping parameter $N \in [5, 10, 20]$, while keeping

Figure 4: (a) A simple Markov reward process from Sutton et al. [2016]. (b) Observed divergence for different sampling distributions $d$ and transition probabilities $p$. (c) Assuming a perfect model and an unbiased data-dependent estimate of $d$ sampled from an instantiation of the environment with $p(S_{t+1} = 1|S_t) = 0.5$, we plot the likelihood of observing divergence as a function of the number of samples used to estimate the empirical distribution $d$.

Figure 5: **Left**: an ablation experiment where we investigate the effect of various settings for the length of the multi-step bootstrapped targets. **Center**: an ablation experiment where we compare our variant of Rainbow to performing updates every 4 steps as in the canonical Rainbow DQN. **Center** comparing our data efficient Rainbow DQN with $M = 1, P = 32$ to a different Rainbow DQN which achieves the same $4\times$ increase in the number of transitions sampled from replay, by increasing the batch size instead ($M = 4, P = 128$).

the update frequency fixed ($M = 1$). Consistently with our expectations a bootstrapping length of $N = 5$ resulted in much worse performance, although our such variant of Rainbow DQN still achieved results comparable to those of SimPLe. Both $N = 10$ and $N = 20$ resulted in good performance, with the difference between the two not found to be statistically significant (under a Welch's test applied to the 5 replicas of each hyper-parameter evaluation, with significance level of 0.1). In Figure 5b we show the effect of varying the frequency of the updates $M \in [1, 4]$, while keeping the number of steps before bootstrapping fixed ($N = 20$); The agent which performs updates on each step performed much better, and the gap in performance was larger then the gap observed when varying the bootstrap parameter $N$. Finally, in Figure 5c we report an additional experiment where we compare our variant of Rainbow DQN with $M = 1, P = 32$ to a different Rainbow DQN which achieves the same $4\times$ increase in the number of transitions sampled from replay, by increasing the batch size instead ($M = 4, P = 128$). The performance was significantly lower.

# E  Table of results

In Table 1 we report, for each of the 26 Atari game used by Kaiser et al. [2019] in their experiments, the mean episode return, at the end of training, of both SimPLe and our data efficient variant of Rainbow. On each score we mark in bold the best performing among the two agents. We also report the reference human and random scores that were used to normalize the scores in all learning curves.

| Game | Human | Random | SimPLe | Rainbow |
|---|---|---|---|---|
| alien | 7127.7 | 227.8 | 405.2 | **739.9** |
| amidar | 1719.5 | 5.8 | 88.0 | **188.6** |
| assault | 742.0 | 222.4 | 369.3 | **431.2** |
| asterix | 8503.3 | 210.0 | **1089.5** | 470.8 |
| bank_heist | 753.1 | 14.2 | 8.2 | **51.0** |
| battle_zone | 37187.5 | 2360.0 | 5184.4 | **10124.6** |
| boxing | 12.1 | 0.1 | **9.1** | 0.2 |
| breakout | 30.5 | 1.7 | **12.7** | 1.9 |
| chopper_command | 7387.8 | 811.0 | **1246.9** | 861.8 |
| crazy_climber | 35829.4 | 10780.5 | **39827.8** | 16185.3 |
| demon_attack | 1971.0 | 152.1 | 169.5 | **508.0** |
| freeway | 29.6 | 0.0 | 20.3 | **27.9** |
| frostbite | 4334.7 | 65.2 | 254.7 | **866.8** |
| gopher | 2412.5 | 257.6 | **771.0** | 349.5 |
| hero | 30826.4 | 1027.0 | 1295.1 | **6857.0** |
| jamesbond | 302.8 | 29.0 | 125.3 | **301.6** |
| kangaroo | 3035.0 | 52.0 | 323.1 | **779.3** |
| krull | 2665.5 | 1598.0 | **4539.9** | 2851.5 |
| kung_fu_master | 22736.3 | 258.5 | **17257.2** | 14346.1 |
| ms_pacman | 6951.6 | 307.3 | 762.8 | **1204.1** |
| pong | 14.6 | -20.7 | **5.2** | -19.3 |
| private_eye | 69571.3 | 24.9 | 58.3 | **97.8** |
| qbert | 13455.0 | 163.9 | 559.8 | **1152.9** |
| road_runner | 7845.0 | 11.5 | 5169.4 | **9600.0** |
| seaquest | 42054.7 | 68.4 | **370.9** | 354.1 |
| up_n_down | 11693.2 | 533.4 | 2152.6 | **2877.4** |

Table 1: Mean episode returns of Human, Random, SimPLe and Rainbow agents, on each of 26 Atari games. The Rainbow results are measured at the end of training and averaged across 5 seeds; the results for SimPLe are taken from Kaiser et al. [2019]. On each game we mark as bold the higher score among SimPLe and Rainbow.

# F Atari hyper-parameters

In Table 2 we report, for completeness and ease of reproducibility, the hyper-parameter settings used by the canonical Rainbow DQN agent, as well as the hyper-parameters that differ in our data efficient variation. renewcommand11.2

| Hyper-parameter | setting (for both variations) |
|---|---|
| Grey-scaling | True |
| Observation down-sampling | (84, 84) |
| Frames stacked | 4 |
| Action repetitions | 4 |
| Reward clipping | [-1, 1] |
| Terminal on loss of life | True |
| Max frames per episode | 108K |
| Update | Distributional Double Q |
| Target network update period* | every 2000 updates |
| Support of Q-distribution | 51 bins |
| Discount factor | 0.99 |
| Minibatch size | 32 |
| Optimizer | Adam |
| Optimizer: first moment decay | 0.9 |
| Optimizer: second moment decay | 0.999 |
| Optimizer: $\epsilon$ | 0.00015 |
| Max gradient norm | 10 |
| Priority exponent | 0.5 |
| Priority correction** | $0.4 \rightarrow 1$ |
| Hardware | CPU |
| Noisy nets parameter | 0.1 |

| Hyper-parameter | canonical | data-efficient |
|---|---|---|
| Training frames | 200,000,000 | 400,000 |
| Min replay size for sampling | 20,000 | 1600 |
| Memory size | 1,000,000 steps | unbounded |
| Replay period every | 4 steps | 1 steps |
| Multi-step return length | 3 | 20 |
| Q network: channels | 32, 64, 64 | 32, 64 |
| Q network: filter size | $8 \times 8, 4 \times 4, 3 \times 3$ | $5 \times 5, 5 \times 5$ |
| Q network: stride | 4, 2, 1 | 5, 5 |
| Q network: hidden units | 512 | 256 |
| Optimizer: learning rate | 0.0000625 | 0.0001 |

\* The target network update period depends on the number of updates (not frames). This means that this update is more frequent in the data-efficient variant, in terms of frames.
\*\* The priority correction linearly annealed from 0.4 to 1 during training: exponent $= (1 - \eta) \times 0.4 + \eta \times 1.0$, where $\eta = $ current_step/max_steps. For the canonical variant, max_step $= 50M$, for the data-efficient variant max_step $= 100K$

Table 2: The hyper-parameters used by the canonical and the data-efficient variant of the Rainbow DQN agent.