[Reviews · NeurIPS 2019]

Reviewer 1



The paper analyzes when to model-based planning instead of replay-based learning and uses several examples to illustrate the idea and also gives theoretical analyses. This paper is well-written and easy to understand. The analysis is inspiring for other researchers and contributes to the RL fields. Comments after author response: Although the experiments are not extensive and the generality of the claims should be validated further, it's beneficial to solicit the discussion of some fundamental issues in RL, such as MPC and replay memory. It's better to clarify the generality ability of claims before a jump of conclusion.

Reviewer 2



This paper broadly considers the use of a learned parametric model. Through (1) toy examples, (2) theoretical analysis of a Dyna-like algorithm, and (3) a large scale study of sample-efficient model-free RL, it arrives at the conclusion that “using an imperfect (e.g., parametric) model to generate fictional experiences from truly observed states… should probably not result in better learning.” While the individual pieces described above are all valuable, I am not sure this claim is properly qualified. For example: “More generally, if we use a perfect model to generate experiences only from states that were actually observed, the resulting updates would be indistinguishable from doing experience replay. In a sense, replay is a perfect model, albeit only from the states we have observed.” I am not sure this is, as stated, exactly true. If the policy has been trained since the state was put in the replay buffer, or if the policy were stochastic, generating experiences from that state would amount to taking a new action on an old state. A predictive model could perform such a counterfactual query, whereas this is not possible with a replay buffer. (The authors address these counterfactual queries in lines 134-142 in the context of planning from the current state, but these queries are useful even in a non-MPC context.) Similarly, if an environment is stochastic, generating experiences with a parametric model using even observed states and past actions would generate new trajectories. More generally, the question of whether a parametric model can generate data to improve a policy or value function should largely come down to whether the model can generalize better than the value function. In a tabular setting in which a predictive model is based on counts, it would not be expected that a model has much to contribute to learning. However, in continuous settings requiring approximation, a model could plausibly generalize better than a value function because of either (1) the difference in supervision or (2) an inductive bias incorporated into the model. Generic neural networks impose a smoothness bias on trajectories, which may be useful for generalization, but other types of predictive models exist. For example, if the model were a differentiable physics simulator with learned parameters, then it could be expected to generalize much better than a neural network policy, albeit at the cost of added bias. A consideration of such types of models is out of scope for this paper, but since the central claim that a model should not be useful is mostly a statement about expected generalization, it would be useful to mention the types of models being considered explicitly. There are some other experimental ways to tease apart the pieces of this hypothesis. For example, it might be reasonable to expect that a parametric model does not generalize any better than a policy with respect to changing state distributions. However, a model may be able to make accurate predictions on the state distribution of the policy that filled the replay buffer. This would allow the model to “fill in” trajectories from the old policy even if it does not generalize to a new policy. Whether or not this is useful to training a policy would then likely amount to an analysis of sampling error, since it would allow for more accurate expectations to be taken over off-policy state distributions. This could be tested by making the fairly reasonable assumption that the model is accurate, but only on a stale policy, and testing whether this extra off-policy data helps. The comparison to SimPLE is valuable in its own right, since it is good to know that a model-free method designed for sample efficiency can outperform a model-based method even in the low data regime in Atari. It is a bit hard to know how to interpret this, given that neither approach seems to work that well with the amount of data given (human-randomized normalized returns of 0.1), but the result is useful as a point of comparison. (It is difficult to criticize when SimPLE is the state of the art for model-based RL on Atari.) Overall, I think a general discussion of this sort about the role of predictive models in RL could be useful to the community. While I have some concerns about the context in which the general conclusions of this paper are presented, I expect that these can be addressed and would be willing to increase my score. ---- Edit after author response ---- Thanks to the authors for responding to my concerns about the paper. The experiment with quadratic dynamics does address my major reservation reasonably, and I would encourage the authors to include this experiment in the final version. It would also be worth including a short discussion about the types of predictive models that may be learned, including situations in which inductive biases could make model learning easier than value function learning, when setting up the scope for the rest of the results.

Reviewer 3



The paper investigates the use of parametric models and relates it to the use of non-parametric experience replay. Training an agent on the replay buffer is seen a planning in the sense that it allows the agent to improve using additional computation. Furthermore, the paper views replay-based algorithms and model-based algorithms under a unified algorithm and compares their performance in a fair setting to show that replay-based algorithms can perform as well or even better than model-based algorithms in certain conditions. In the popular Atari benchmark, It is shown that the replay-based algorithm Rainbow DQN is able to achieve better performance than a recent state-of-the-art model-based approach called SimPLe, using less data and computation. Since the experimental results suggest that parametric models are not always necessary, the paper further investigates the necessity of parametric models. It is shown that planning in Dyna-style algorithms can easily lead to catastrophic learning updates. Alternatively, the paper suggests to instead use backward models which predict previous state and action from the current state and reward. This is beneficial because poor forward models lead to erroneous updates in an observed state while poor backward models only leads to erroneous updates in imaginary states. However, it might be more challenging to learn backward models due to their multi-modal nature even in simple deterministic environments. The paper is very well written and easy to follow. The conceptual and experimental contributions of the paper are significant and future works are likely to build upon them. I recommend acceptance of the paper. Post-rebuttal comments: Thanks for the clarifications and updates.

[Author Response · NeurIPS 2019]

# Author response for 'When to use parametric models in reinforcement learning?'

We sincerely appreciate the reviewers' time and effort to provide useful and insightful reviews, and in this case of course especially for their helpful comments and question about our paper.

## On code and reproducibility

We took care that our experiments are reproducible, and we will recheck this carefully on acceptance. We are happy to report that our main experiment has already been successfully reproduced by others, without access to our code, which is a good validation that the paper contained sufficient details. We of course appreciate that releasing code can help speed up research (and can sometimes help clarify important details), and intend to release accompanying code, and we also think it is important to make sure the paper itself contains sufficient detail to fully reproduce the results.

## On model benefits

One reviewer raises interesting and insightful points about the usefulness of models that generalise better than a policy or value. We agree that this is an important aspect that is worthwhile to discuss in some detail.

An important, and perhaps obvious, conclusion is: it is not just important how accurate the model is (though this clearly matters), but also the model is used. We agree with the reviewer that models that generalise better than a policy or value seem especially interesting, when these are attainable. Interestingly if we then use the model in the same way as we could use replay (as opposed to, for instance, using it to plan forward for behaviour), then only accuracy does not always suffice, even in such benign settings.

It might be interesting to explain an experiment that we conducted, but that did not make it into the paper (yet). We set up an experiment in which the true transition dynamics were quadratic, and the model was also chosen to be quadratic. Non-surprisingly, the model very quickly learnt to match the true dynamics, essentially perfectly. To our initial surprise, the forward Dyna algorithm didn't perform well even in this presumably benign setting—it performed far worse than the replay-based algorithm and the parameters of the value function would often diverge. We first suspected a bug but careful examination revealed this was, instead, a failure of the sort that is now discussed in Section 3. In the appendix we chose a simpler (two-state) example to concretely illustrate the more general theory around this failure, but perhaps it is useful to include this quadratic example in the paper as well, as another demonstration that even a model that generalises perfectly can fail if not used with care.

To be clear, of course we agree that a perfect model, if available, can help attain performance that should surpass that attained from using replay instead. We are just pointing out that using the model only to generate fictional data in the same way (and from the same states) as replay would may not be the easiest way to benefit from an imprecise model.

Of course, it can be hard to know a priori when a model will be accurate enough to rely on, especially for longer trajectories where compounding model errors can be a problem [cf., e.g., Talvitie, 2014, 2017, Asadi et al., 2019]. To quote Vladimir Vapnik: *one should (...) never solve a more general problem as an intermediate step* [Vapnik, 1998, Section 0.9]. Any statement of such generality comes with caveats, but it is interesting to consider in this context: when we use replay we at least know that the data is real, and we do not have to question its accuracy.

We agree that it is important to state our findings as clearly as possible (and appropriately scoped, to avoid hinting toward unwarranted overly general conclusions) and we intend to carefully keep polishing the writing to make the paper as clear as possible.

Relatedly we are also considering including further empirical results (including, though not exclusively, more at-scale results, e.g., with backward planning on Atari, as well as perhaps including the experiment with quadratic dynamics described above) to help further elucidate our main points and augment the experiments currently in the paper.

# References

K. Asadi, D. Misra, S. Kim, and M. L. Littman. Combating the compounding-error problem with a multi-step model. *CoRR*, abs/1905.13320, 2019.

E. Talvitie. Model regularization for stable sample rollouts. In *UAI*, pages 780–789, 2014.

E. Talvitie. Self-correcting models for model-based reinforcement learning. In *Thirty-First AAAI Conference on Artificial Intelligence*, 2017.

V. Vapnik. *Statistical learning theory*. Wiley, 1998.


[Meta-Review · NeurIPS 2019]

There's consensus that this is a well written paper that offers some useful insights about the pros and cons of model-based RL vs. model-free RL with replay buffers. This is an important topic and this paper has the potential to make significant impact. However, the authors are urged to be careful about not making overly general conclusions in the final version of the paper, as this was a concern of one reviewer. Even the title may be too general. Strong paper overall though.